# A Threshold Helium Leakage Detection Switch with Ultra Low Power Operation

**DOI:** 10.3390/s23084019

**Published:** 2023-04-15

**Authors:** Sulaiman Mohaidat, Fadi Alsaleem

**Affiliations:** 1Department of Mechanical and Materials Engineering, University of Nebraska-Lincoln, Lincoln, NE 68588, USA; smohaidat2@huskers.unl.edu; 2Department of Architectural Engineering, Durham School of Architectural Engineering and Construction, University of Nebraska-Lincoln, Omaha, NE 68182, USA

**Keywords:** electrostatic MEMS, leakage detection, helium sensor, dielectric constant, dry cask nuclear waste storage

## Abstract

Detecting helium leakage is important in many applications, such as in dry cask nuclear waste storage systems. This work develops a helium detection system based on the relative permittivity (dielectric constant) difference between air and helium. This difference changes the status of an electrostatic microelectromechanical system (MEMS) switch. The switch is a capacitive-based device and requires a very negligible amount of power. Exciting the switch’s electrical resonance enhances the MEMS switch sensitivity to detect low helium concentration. This work simulates two different MEMS switch configurations: a cantilever-based MEMS modeled as a single-degree-freedom model and a clamped-clamped beam MEMS molded using the COMSOL Multiphysics finite-element software. While both configurations demonstrate the switch’s simple operation concept, the clamped-clamped beam was selected for detailed parametric characterization due to its comprehensive modeling approach. The beam detects at least 5% helium concentration levels when excited at 3.8 MHz, near electrical resonance. The switch performance decreases at lower excitation frequencies or increases the circuit resistance. The MEMS sensor detection level was relatively immune to beam thickness and parasitic capacitance changes. However, higher parasitic capacitance increases the switch’s susceptibility to errors, fluctuations, and uncertainties.

## 1. Introduction

A dry cask is a structure that stores nuclear waste inside a sealed metallic cylinder hosting the spent fuel. The canister is filled with helium to cool the fuel by convection and provides the inert conditions to protect the fuel cladding (zirconium metal alloy) from re-oxidation, thus protecting the whole structure [1]. Because maintaining the inert environment is essential, several methods were developed to detect helium leakage from the canister. For example, Takeda developed several thermal methods to detect helium leaking in dry casks, including monitoring the temperature at several locations along the structure. These include the top and bottom of the canister and the air at the inlet of the canister [2]. However, this approach is limited due to the indirect detection of helium leakage and the susceptibility to errors associated with temperature measurement.

Outside the application of helium in nuclear applications, helium sensing, in general, is still challenging. Helium is an inert gas, which is stable and chemically non-reacting, making it challenging to sense using chemical methods [3]. Thus, other non-traditional methods were proposed in the literature to measure helium. For example, phase interference surface plasmon resonance that depends on the gas refractive index was used to measure helium [3]. The detection of helium in the air was also proposed and accomplished by measuring the changes in resonance frequency based on gas mixtures in a tube [4]. Hydrated vanadium pentoxide V_2_O_5_·1.6 H_2_O nanostars were also proposed as a potential helium-sensing material at room temperature because their resistance decreases with the presence of helium [5]. Yttrium functionalized open-edge boron nitride has been studied as a helium sensing material [6]. Dong et al. [7] utilized the change in low-field emission of multi-walled carbon nanotubes to detect helium at low pressure. Mahdavifar [8] et al. developed a helium sensor based on the change in the electrical resistance of doped polysilicon with a sensitivity of 0.34 mΩ/ppm. The voltage gain of a relatively large-size triboelectric nanogenerator was used in [9] to detect inert gases, including helium and argon, with a 180% increase in the open circuit gain for helium ionization. The acoustic techniques that depend on measuring the change in thermal conductivity and speed of sound caused by helium/air mixing were developed by Sheen et al. [10]. Table 1 provides a summary of recent helium detection sensors. While promising, these approaches require complex data processing equipment, advanced experimental materials, and relatively large sensor sizes. To overcome these challenges, in this work, we explore the use of electrostatic microelectromechanical systems (MEMS) for detecting helium.

MEMS has recently been proposed in the literature for sensing helium [11]. It offers the possibility of detecting helium using a tiny sensor size. However, the current MEMS helium sensor technology is based on the higher thermal conductivity of helium compared with other gases and requires heating the MEMS sensing element. This requirement might complicate the MEMS design and operation. In this paper, we demonstrate a new concept of a MEMS helium sensor that is based on changes in the electrical property of air due to the presence of helium and does not require thermal actuation. In addition, it allows utilizing the MEMS dynamic instability due to pull-in [12] to achieve helium detection without the need for complex signal conditioning and interface circuitry.

## 2. Method

Electrostatic MEMS can be used for capacitance sensing. In this case, assuming a parallel plate configuration, sensing depends on measuring the change in capacitance between the fixed and movable plates of a parallel plate capacitor. The capacitance in this case is:(1)$$C=\frac{\varepsilon_{r}\varepsilon_{o}\;A}{g}$$
where *A* is the area of the electrodes, *g* is the gap between the electrodes, and $\varepsilon_{r},\varepsilon_{o}$ are the relative permittivity (or dielectric constant) for the medium between the electrodes and permittivity of space, respectively. Capacitive MEMS can also be used as electrostatic actuators, the capacitor’s electric field generates an electrostatic force which will cause the movable plate to move toward the fixed plate [13].

The dielectric constant ($\varepsilon_{r}$) of a material depends on its electric dipole polarizability, which is a material property that depicts the behavior of molecules and atoms when subjected to an external electric field [14]. This behavior is known as the molar polarizability (℘) of a material and can be calculated from the equation [15]:(2)$$℘=\frac{\varepsilon_{r}-1}{\varepsilon_{r}+1}\;\frac{1}{\rho }=A_{\epsilon }\left[1+b\left(T\right)\rho +c\left(T\right)\rho^{2}+\ldots \right]$$
where the coefficients $A_{\epsilon }$, $b\left(T\right)$, and $c\left(T\right)$ are calculated ab initio. The dielectric constant of air is 1.00059 [16] and the dielectric constant of helium is 1.0000665 [17], which is less than other common gases in the air. Thus, helium leakage to air results in a lower air dielectric constant and a lower MEMS capacitance (*c*). Based on this observation, capacitive-based sensors were shown in the literature to measure changes in air permittivity due to different gases, including helium [15]. However, these sensors require sophisticated sensor readout and complex design.

To overcome the above challenge, we explore using the complex nonlinear dynamics of a simple capacitive parallel-plate MEMS device to realize a novel threshold helium sensor (THS). Specifically, the nonlinear dynamics of electromechanical coupling of electrostatic MEMS actuators causes the phenomena known as pull-in instability, which happens when the electrostatic force (due to an applied voltage) more quickly increases than the spring force leading to the collapse of the moving plate [12,13]. The pull-in voltage depends on the system’s capacitance and hence the MEMS structure’s dielectric constant, a function of the helium concentration.

### THS Working Principle

The THS sensor indicates possible leakage when the helium concentration exceeds a threshold value instead of measuring the concentration. A similar concept has been realized for acceleration applications [8] (e.g., firing a car’s airbag system when its acceleration exceeds a threshold value) and mass sensing applications [18]. To realize the THS, a bias voltage is applied to a MEMS microstructure, such as a clamped-clamped beam (Figure 1a), to cause a pull-in for the structure. This leads to the collapse of the structure. In Figure 1b, by tuning the switch with a specific bias pull-in voltage, the THS sensor turns off (pull-out) only when the helium concentration exceeds a predetermined threshold. For example, as shown in Figure 1b, the THS will detect a lower HE concentration at S1 than at S2 by providing a higher bias voltage. This design produces a tunable helium detection switch. It eliminates the need for an analog sensor with a complex interface (amplification, A/D circuitry, controllers, and decision units) and offers an easy way to tune the threshold value.

Orders of magnitude improvement in the THS sensitivity to helium concentration can be achieved by utilizing the concept of electrical resonance activation in MEMS sensors. In this operation, a small AC voltage signal with a frequency that matches the electrical resonance of the MEMS circuit will significantly increase the MEMS capacitive sensitivity to changes in air permittivity and hence helium concentration. Specifically, operating the MEMS RLC circuit by adding an inductance in series at the resonance frequency ($1/2\;\pi \;\sqrt{L\;C\;}\;$(Hz)) will lead to voltage amplification. This voltage amplification is dependent on the MEMS capacitance and other circuit parameters as described by this equation [19]:(3)$$\left|\frac{V_{ac}}{V_{in}}\right|=\frac{1}{\sqrt{{\left(\left(2\pi f\right)RC\right)}^{2}+{\left({\left(2\pi f\right)}^{2}LC-1\right)}^{2}}}$$
where $V_{ac}$ is the amplified voltage and $V_{in}$ is the input voltage.

## 3. Results

### 3.1. Commercial of the Shelf MEMS Sensor Modeled as a Single Degree of Freedom Model

As a proof concept, we first used a model for a cantilever beam with a proof mass pinned from the middle, as shown in Figure 2a. This device, produced by Sensata Technologies, is a commercially available capacitive sensor that has been extensively investigated in the literature, and a good match between its simulation results and experimental data was reported [12]. A schematic of the MEMS device is shown in Figure 2b.

The MEMS sensor-proof mass, forming one side of a capacitor electrode, has a length *l* (m), width *b* (m), and thickness *h* (m). An initial gap *d* (m) separates the proof mass from the substrate beneath. The following single-degree freedom model can describe the dynamic response of the beam:(4)$$m_{eff}\ddot{z}\left(t\right)+c\left(z\right)\dot{z}+kz=F_{e}\left(z,t\right)$$
where z  is the deflection, negative upwards and positive downwards (m), the dot operators represent temporal derivatives, t is the time in seconds (s), the effective mass $m_{eff}$ is given by $\frac{k}{\omega_{n}^{2}}\;$, $\omega_{n}\;$ is the natural frequency of the system $\left(\mathrm{rad}/s\right)$, and *F_e_* is the electrostatic force given by:(5)$$F_{e}=\frac{\epsilon \left(He\right)A_{s}{\left[V_{DC}+V_{AC}Cos\left(\Omega t\right)\right]}^{2}}{2{\left(d-z\right)}^{2}}$$
where Ω  represents the AC excitation frequency (rad/s) and $A_{s}$ the area of overlap between the microbeam-proof mass and the substrate beneath it ($m^{2}$).

As a demonstration of the THS operation, Figure 2c shows the normalized MEMS response at different helium concentrations with a bias voltage slightly above the pull-in voltage. The figure shows a minimum detection level threshold of 50,000 PPM can be achieved. While the sensor minimum detection level can be improved by optimizing the MEMS design, electrical resonance activation results in multiple order of magnitude sensitivity improvement (Figure 2d) [19,20]. In this operation, a small AC voltage signal with a frequency that matches the electrical resonance of the MEMS circuit will significantly increase the MEMS capacitive sensitivity to changes in air permittivity. Thus, the commercial capacitive sensor can achieve a minimum sensitivity of up to 300 PPM.

### 3.2. Micro-Continuous Beam Finite Element Model

In the second case study, we present a complete detailed analysis of a continuous micro-beam response to the presented helium sensing concept. The finite element software package COMSOL Multiphysics was used for this study. The electromechanics physics interface was used, which combines electrostatics and solid mechanics and solves the electromechanical forces (by solving the Minkowski electromagnetic stress tensor). The mesh used in this study is physics-controlled tetrahedral mesh with an extremely fine setting as defined by COMSOL Multiphysics.

COMSOL simulations were conducted using the UNL supercomputer cluster CRANE. They use Intel Xeon CPUs, including Xeon E5-2697 V4 and Xeon Gold 6248. Simulations used a variable number of nodes 1–16, each containing 32 cores, with waiting time from several hours up to two weeks and computing times taking up to several days even when using coarse mesh. Most of our simulations were conducted using the time-dependent solver of COMSOL. Output sizes reached more than 100 GB files, especially with long simulation times that passed 100 µs. The default time stepping was used because fixing the time step to the smallest size taken by the solver will make it highly unlikely to finish the simulations in the allowable run time rules.

A polysilicon beam base design (Figure 3) with a width of 300 µm (x-dimension), a depth of 8 µm (y-dimension), and a height of 20 µm (z-dimension) was used for in-plane actuation (negative y-direction). The gap thickness is 10 µm. The dielectric constant of air was approximated to 1.0005, and for full helium was considered 1.0001, the vacuum permittivity was 8.854 × 10^−12^ F·m^−1^, and the nominal capacitance was calculated to be 5.3151 ×10^−15^ F. To account for the parasitic capacitance, 10 times the nominal capacitance was used, leading to a total system capacitance of 5.8466 × 10^−15^ F. In this work, the change in the dielectric constant due to helium presence (from 1.0005–1.0001) was assumed to follow a linear rule of mixtures. Thus, the helium detection level (HDL) is calculated by the following equation:(6)$$\mathrm{HDL}=\frac{\epsilon -1.0005}{1.0001-1.0005}\times 100\%$$

The detected dielectric constant is the maximum dielectric constant less than 1.0005 (full air), at which the beam will not suffer a pull-in event. This study used the dielectric constants of 1.0004, 1.00043, 1.00045, 1.00048, and 1.0005 corresponding to helium detection levels of 25, 17.5, 12.5, 5, and 0% (full air), respectively. While a helium detection level of less than 5% can theoretically be achieved, we limit our study to 5% due to the demanding computation effort. When a DC voltage is ramped from 0 to pull-in voltage, with a course variable voltage step size, the beam displacement response difference between full helium (ε_r_ of 1.0001) and full air (ε_r_ of 1.0005) is not noticeable, as shown in Figure 4. In this simulation, we used the stationary solver of COMSOL Multiphysics. While a noticeable detection level can be achieved by lowering the voltage step size, similar to Figure 2, this will not be significant. Moreover, lowering the step voltage requires significant computing power and may last weeks, even using a supercomputer.

Next, we added a series resistance of 40 Ω and an inductance of 30 mH to form an RLC circuit with a resonance frequency of 3.8 MHz. A small amplitude AC signal that matches the MEMS electrical resonance frequency was applied. The amplification Equation (3) was used to calculate the amplitude of the terminal voltage (beam voltage) at the actuation AC frequency. By triggering the MEMS electrical resonance, the MEMS electrostatic sensitivity to the input voltage and hence to the dielectric constant of the medium (gas mixture of air and helium in our case) increases. Figure 5 validates this point. Figure 5a shows the beam response due to different helium concentrations, represented by the different dielectric constants. Figure 5a shows that with a dielectric constant of 1.0005 (full air), the electrostatic force was enough to cause the beam to pull in (switch on). However, by reducing the dielectric constant to 1.0004 (helium concentration of 25%) or less, the beam gained stability and escaped the pull-in (switch off). Moreover, Figure 5b confirms the sensitivity of the terminal voltage (effective voltage across the MEMS) to the dielectric constant values. In conclusion, compared with Figure 4, Figure 5 shows the effective use of MEMS electrical resonance to increase its sensitivity to helium detection. Another advantage of utilizing electrical resonance, as reported in [19,20], is the massive reduction of the actuation voltage to achieve pull-in.

Figure 6 pushes the minimum detection level of helium lower by simulating a dielectric constant of 1.00048, which deviates only by 0.00002 from the dielectric constant of air. Surprisingly, the figure shows that the beam still escapes the pull-in even with this minimal dielectric constant reduction.

Next, we will discuss the effect of the AC excitation frequency on the sensor sensitivity. Table 2 summarizes the different excitation frequency values and their effect on the minimum helium detection level. The table shows that with the reduction of the excitation frequency from the electrical resonance value, two behaviors are observed: (1) the increase in the required input voltage to achieve pull-in for full air and (2) and the reduction in helium detection performance (i.e., the increased in the minimum detection level). For example, at 3.78 MHz frequency, an input voltage of 7.479 Volt is required to achieve pull-in at full air. Moreover, the helium detection level increased to 25% (Figure 7). This behavior can be attributed to the reduced amplification of the input voltage; hence, less sensitivity to the ε_r_ value, as seen in Figure 8. Table 2 also shows that the minimum helium detection level increased to 50% (ε_r_ = 1.0003) at 3.77 MHz. This further emphasizes the importance of electrical resonance physics.

In summary, Table 2 shows that the minimum helium detection level increases when the excitation frequency departs from the electrical resonance frequency, degrading the sensor’s performance. Using electrical resonance, the magnitude of voltage amplification depends on the dielectric constant value and the excitation frequency, achieving the max amplification at the electrical resonance frequency. Thus, the sensor sensitivity to changes in helium is reduced as the excitation frequency is moved away from the electrical resonance frequency.

One of the critical parameters in the RLC circuit is the resistor, which acts as the electrical damping element. Increasing the resistance value while keeping the excitation frequency at 3.8 MHz increases the input voltage and reduces the helium detection performance, as shown in Table 3. The table shows that increasing resistance up to 100 Ω, the detection level remains at 5%. However, the minimum detection increases to 25% for 1 kΩ. This increase can be explained by plotting the terminal voltage variation in Figure 9 versus the dielectric constant for 40 Ω and 1 kΩ resistances. Figure 9 shows that terminal voltage variation at higher dielectric constant values is almost saturated for the high resistance value (1 kΩ). Thus, a much lower value of ε_r_ = 1.0004 is needed to produce a noticeable variation in the terminal voltage compared with the 40 Ω resistance (ε_r_ =1.00048).

Any MEMS electrical circuit may suffer parasitic capacitance due to wiring and the different circuit parts. Next, in Figure 10, we investigate the parasitic capacitance effect on MEMS detection sensitivity. Figure 10 shows that as the parasitic capacitance increases, the total system capacitance increases, and hence the electrical resonance frequency decreases (Figure 10a). This increases the input voltage while matching the excitation frequency to the new electrical resonance frequency (Figure 10b). Despite these changes, our simulations reveal that the minimum possible helium detection level remains at 5%. However, accepting a large parasitic capacitance in the MEMS circuit design may come at the cost of decreasing the sensor stability against noise and environmental conditions.

To elaborate more on this, in Figure 11, we compare the maximum beam displacement and terminal voltage at 5% helium (corresponding to ε_r_ = 1.00048) to the last stable displacement and terminal voltage before pull-in at full air, as we vary the parasitic capacitance. The plots in Figure 11 show that while there is still a clear pull-in (full air) versus no pull-in (with 5% helium) at all parasitic capacitance levels, the difference in displacements and terminals voltages decreases with increased parasitic capacitance. Theoretically, the big difference at lower patristics capacitance means the sensor may be able to detect a lower level of Helium than 5%. However, in this study, as we limited it to 5%, this big difference translates to more stability in the system response to the noise.

Finally, in Figure 12, we show the terminal voltage amplitude at different dielectric constants for the parasitic capacitance of 10, 100, and 1000. The figure confirms the voltage reduction across the full range of dielectric constant as the parasitic capacitance increases. The relationship becomes almost linear at 1000 PCF, explaining the behavior observed in Figure 11a,b.

The effect of beam thickness on the sensing performance is discussed at the end. The beam thickness varies from 6 to 20 µm while keeping other parameters unchanged, as shown in Figure 13. The figure shows that the input voltage amplitude increases as the beam thickness increases, increasing the beam’s mechanical stiffness (Figure 13a). Figure 13b shows that, despite the voltage reduction, there is almost negligible impact on the beam displacement difference between full air and 5% helium concentration.

It is worth mentioning this work has some limitations. For example, the model assumes an ideal inductance, which ignores any internal resistance that may impact the electrical resistance amplification and reduces the sensor sensitivity. Currently, there are efforts to create an experimental set-up to validate some of the simulations presented in this paper.

Finally, to check the effect of temperature on the sensor’s performance, we performed a simulation for a cantilever beam at full air condition, as shown in Figure 14. The fixed end of the cantilever beam had a temperature of 293.15 K, and the other boundaries varied from 239.15 to 493.15 K at 40 K steps. The simulation showed no appreciable difference in displacement as a function of temperature at a constant excitation voltage. This implies that temperature has less impact on these capacitive systems; hence, the presented gas-sensing method is immune to temperature fluctuations. Similar findings on the minimum effect of temperature variation on the MEMS dielectric constant measurement systems in air and with the presence of helium and other gases were presented in [21].

## 4. Conclusions

The ability to detect the presence of helium in the air was computationally demonstrated using two approaches, a lumped mass model approach for a cantilever beam and an extremely fine mesh model in COMSOL Multiphysics using the electromechanics physics interface for a clamped-clamped micro-beam. For both approaches, the difference in the dielectric constant was the basis for detection using an RLC circuit model excited at its electrical resonance frequency. This leads to a dependence of the amplified input voltage on the value of the dielectric constant of the air/helium mixture. The changes in the amplified voltages are tuned to trigger a mechanical switch action for the micro-structures when helium exceeds a specific threshold. This approach eliminates any need for power-hungry conventional sensing conditioning circuits.

The COMSOL, due to its accuracy, was selected for a detailed analysis. The main performance parameter is the helium detection level (HDL) represented by a percentage, with a higher detection level meaning a reduction in the sensor’s performance. An HDL level of 5% is considered the best performance for this study. Using the electrical resonance concept, the ability to detect a 5% helium presence corresponding to a dielectric constant of 1.00048 was demonstrated. The departure from the electrical resonance frequency increased the detection level from 5% (3.8 MHz) to 50% (3.77 MHz). The system is tunable by controlling the input voltage amplitude and frequency, which can account for variations in less controllable parameters such as parasitic capacitance. The parasitic capacitance can potentially degrade the sensor’s performance, especially at higher values. The system’s resistance increased detection levels, especially at values higher than 100 Ω. The system achieved the same sensing capability (but at increased voltage) with increased thickness, which means the design is relatively tolerant to the difficulties and uncertainties associated with fabrications. Simulations also showed the design is immune to temperature fluctuations.

## Figures and Tables

**Figure 1 sensors-23-04019-f001:**
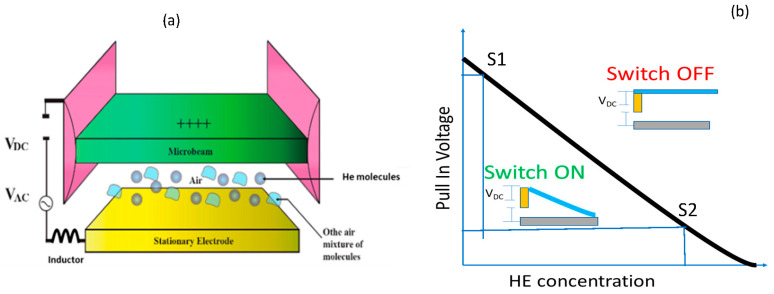
The operation concept of the proposed helium switch. (**a**) A schematic view of a parallel plate cantilever capacitor and (**b**) the switch operation curve, any point above the curve leads to the switch opening a circuit. The switch can be tuned to be triggered OFF at different helium concentrations by changing the bias voltage.

**Figure 2 sensors-23-04019-f002:**
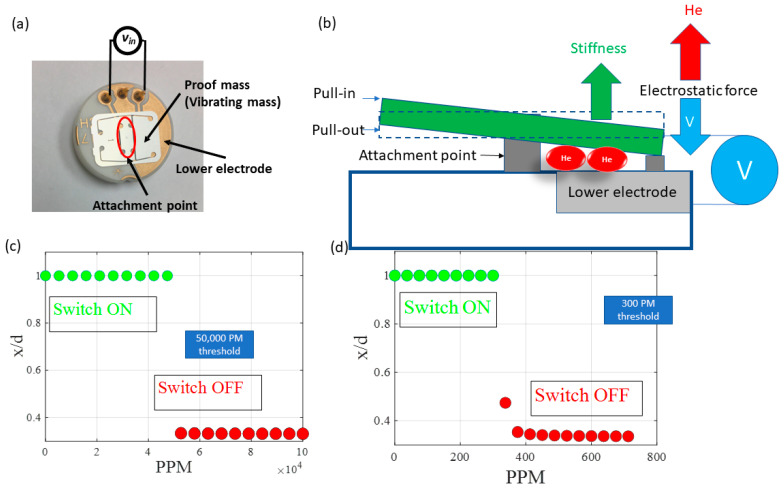
(**a**) An image of the MEMS sensor and (**b**) a schematic. The sensor response at different helium concentration levels (**c**) using the typical DC bias operation and (**d**) adding a small AC voltage signal that activates the electrical resonance circuit formed by the MEMS capacitance.

**Figure 3 sensors-23-04019-f003:**
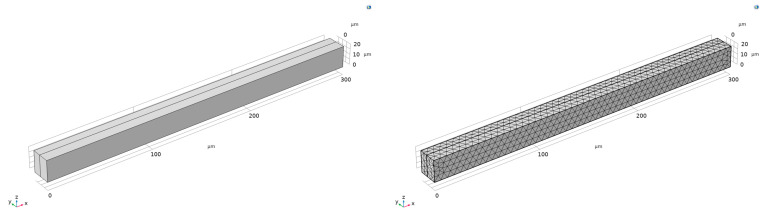
The base beam design geometry of the FE COMSOL model used in the simulation with a mesh sample.

**Figure 4 sensors-23-04019-f004:**
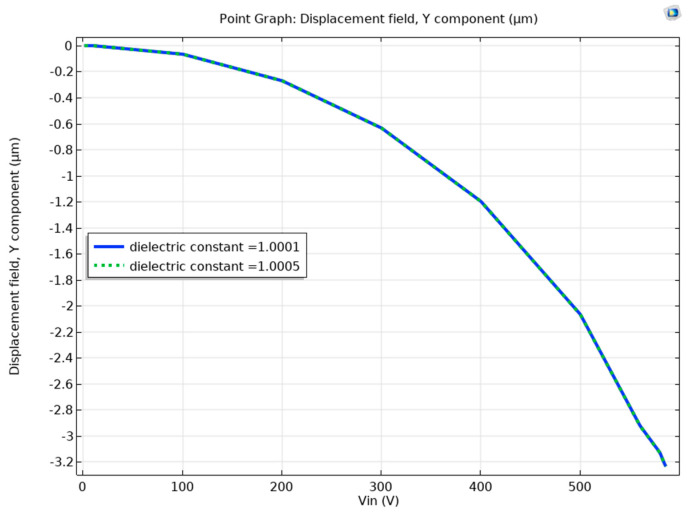
Displacement behavior of the tip of the beam without electrical resonance. Subjected to DC voltage ramping (1–586 V) which showed no visible displacement difference at *ε* of 1.0001 and 1.0005.

**Figure 5 sensors-23-04019-f005:**
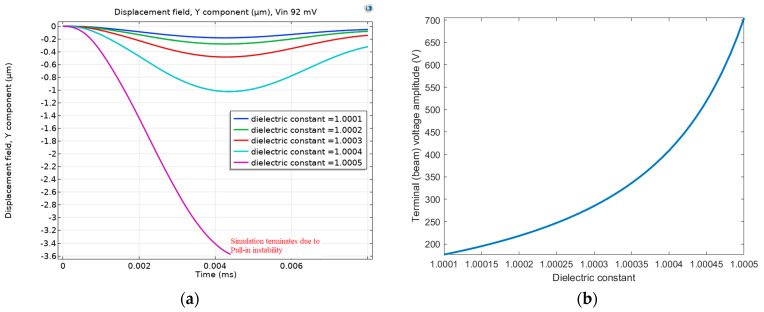
(**a**) The displacement at resonance excitation at 3.8 MHz with 92 mV input voltage. (**b**) The voltage amplification amplitude dependence on the dielectric constant shows the effect of electrical resonance excitation (3.8 MHz).

**Figure 6 sensors-23-04019-f006:**
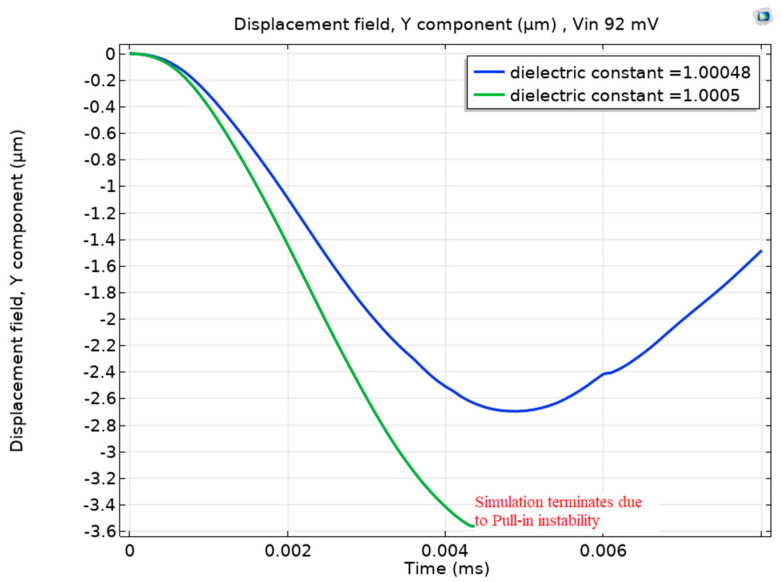
The detection of 5% helium (ε_r_ = 1.00048) at 3.8 MHz (while the pull-in event happened at 0% helium (ε_r_ = 1.0005) condition).

**Figure 7 sensors-23-04019-f007:**
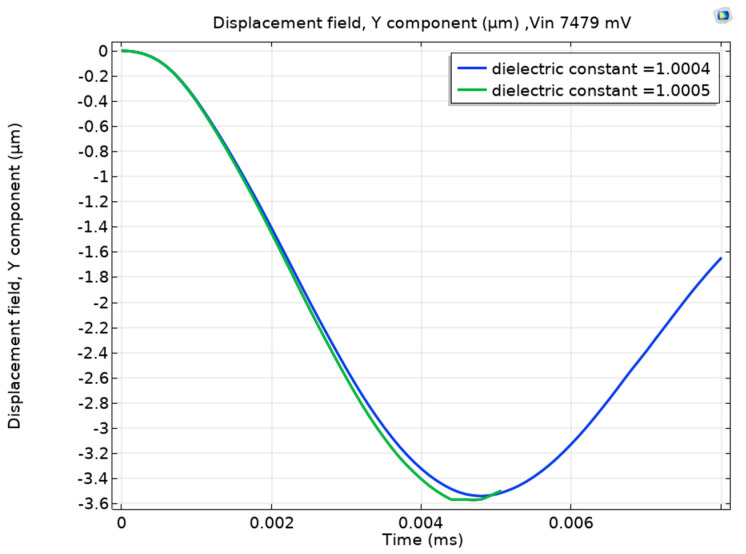
Helium detection at 3.78 MHz; it is noticeable that the detection level increased to 25%, degrading the sensor’s performance.

**Figure 8 sensors-23-04019-f008:**
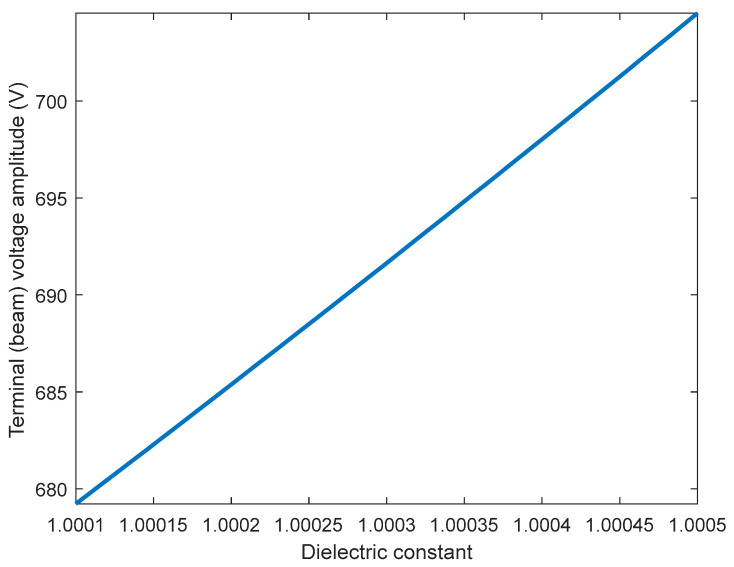
The relationship between the dielectric constant and the terminal voltage amplitude at 3.78 MHz, the reduced voltage range based on ε reduced the sensor’s performance.

**Figure 9 sensors-23-04019-f009:**
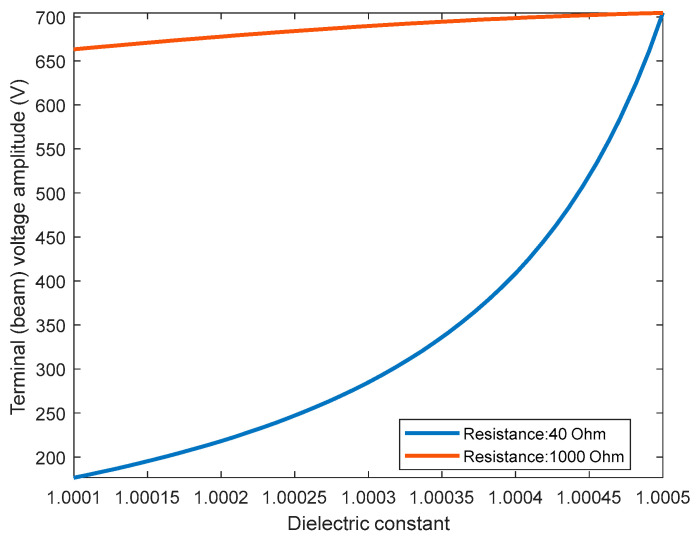
The effect of the resistance on the amplitude of the terminal voltage as a function of the dielectric constant.

**Figure 10 sensors-23-04019-f010:**
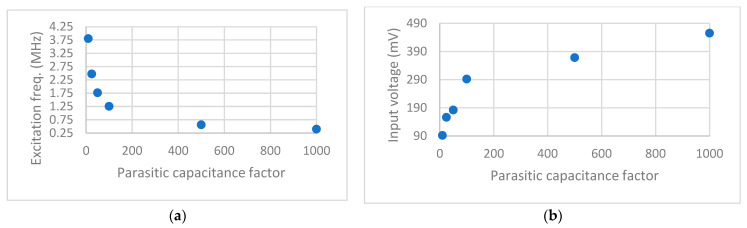
Changes in system parameters due to increased parasitic capacitance (parasitic capacitance factor), (**a**) decreased excitation frequency, and (**b**) increased input voltage.

**Figure 11 sensors-23-04019-f011:**
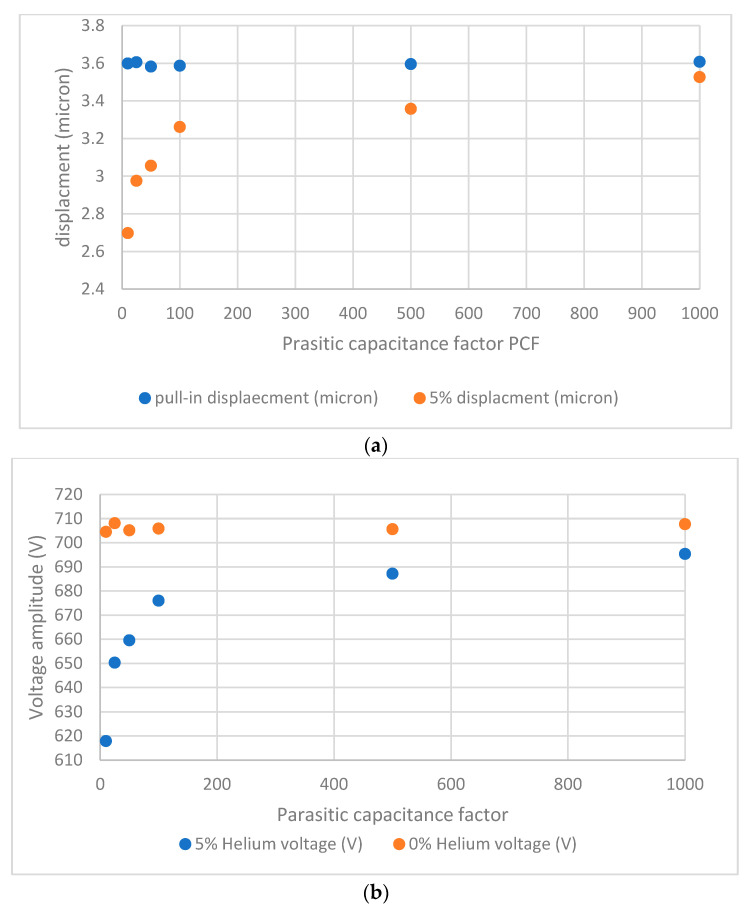
Beam displacement and (**a**) voltage amplification at (**b**) 0 and 5% helium levels as a function of parasitic capacitance factor.

**Figure 12 sensors-23-04019-f012:**
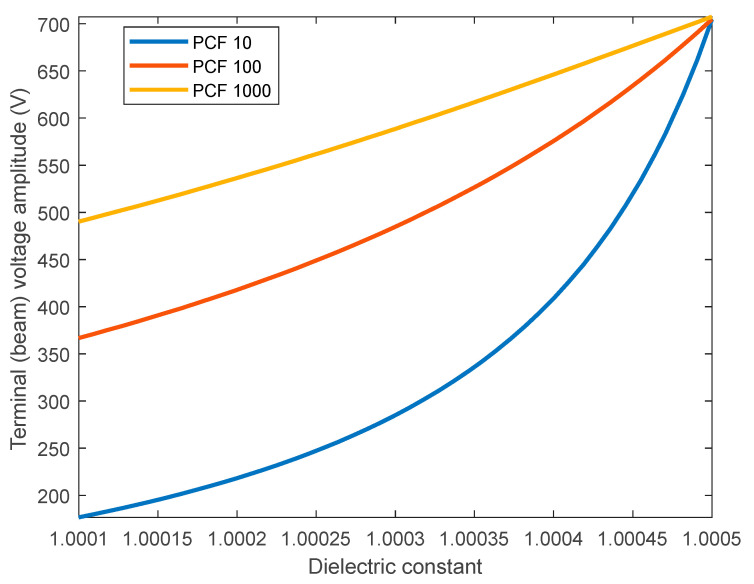
The effect of parasitic capacitance on the voltage amplitude dependence on dielectric constant.

**Figure 13 sensors-23-04019-f013:**
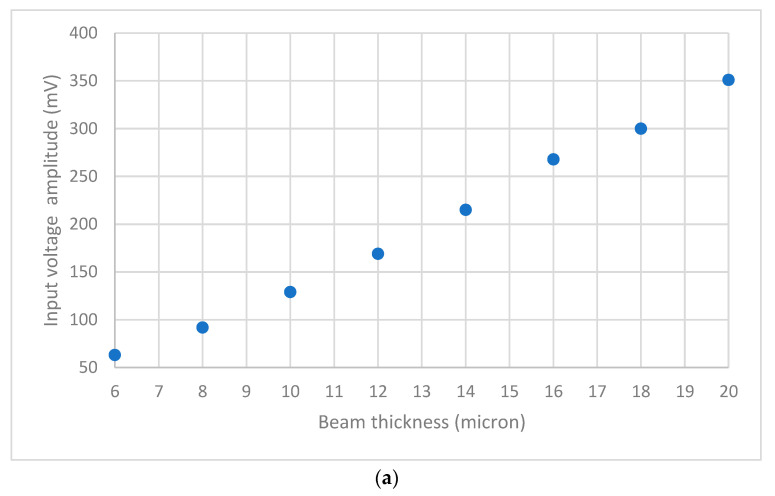
(**a**) Effect of thickness on the input voltage amplitude to the sensor at the same electrical parameters. (**b**) Displacement at 0 and 5% helium levels based on the thickness of the beam.

**Figure 14 sensors-23-04019-f014:**
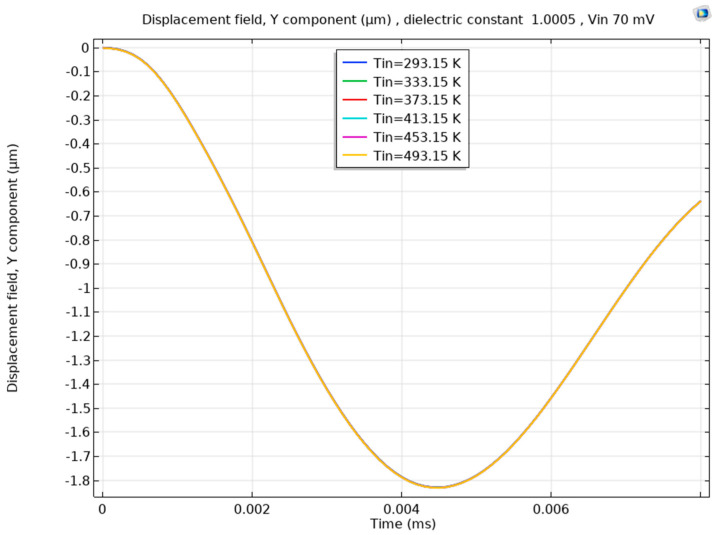
The effect of temperature change on the displacement of the beam at air-only conditions. The figure shows the beam deflection is almost the same. Hence, changes in the die electric constant are minimum against a significant temperature variation.

**Table 1 sensors-23-04019-t001:** Summary of recent helium detection sensors.

Reference	Method	Detection Performance
[3]	Surface plasmon resonance	100% He
[4]	Resonance frequency change	90%
[6]	Band gap change	NA
[7]	Carbon nanotubes emissions	10^−8^ Pa·L/s
[8]	Thermal conductivity	700 PPM
[9]	Open circuit voltage of helium ionization	53 PPM
[10]	Thermal conductivity (acoustic)	10^−4^ cm^3^/s
[10]	Speed of sound	10^−5^ cm^3^/s
[11]	Thermal conductivity	5%

**Table 2 sensors-23-04019-t002:** The effect of excitation frequency on helium detection level and input voltage.

Frequency (MHz)	Input Voltage (mV) to Achieve Pull at Full Air	Helium Detection Level (%)
3.8	92	5
3.79	3787	12.5
3.78999	3790	17.5
3.78	7479	25
3.77	11,163	50

**Table 3 sensors-23-04019-t003:** The effect of the resistance on the input voltage and helium detection level.

Resistance (Ω)	Input Voltage (mV)	Helium Detection Level (%)
40	92	5
50	97	5
100	129	5
250	260	12.5
500	499	12.5
750	743	17.5
1000	987	25

## Data Availability

Not applicable.

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
