# Peer review of "A Threshold Helium Leakage Detection Switch with Ultra Low Power Operation"

_sensors, 2023, doi:10.3390/s23084019_

Round 1

Reviewer 1 Report

The manuscript proposed a Helium concentration MEMS sensor structure based on a cantilever capacitor, using COMSOL finite-element simulation to validate the design. The quality of the figures can be improved before acceptance, i.e., enlarging the font size inside the figures.

Author Response

We value the reviewer's comment. We have revised most of the figures to be clear 

Reviewer 2 Report

Title: A Threshold Helium Leakage Detection Switch with Ultra Low Power Operation

Manuscript ID: sensors-2349595

In this report, the authors have delivered a Helium detection system depending on the relative permittivity (dielectric constant) variance between air and Helium. Subsequently, this variation alters the microelectromechanical system (MEMS) switch state. The work simulates two different MEMS switch configurations. The detection level is 5% helium at 3.8 MHz excitation.

In general, the manuscript is fine, few minor comments are needed

-        Introduce a table of comparison between the detection limit for the proposed work and other research reports

-        Add more recent research articles to the introduction to enrich the introductory part.

-        Provide better figures, particularly for fig. 5, 8,10, and 11.

-        Provide more explanation on the enhancement of the detection limit when departing from the resonance frequency.

-        Check abbreviations.

-        Revise equations order.

The reviewer

04042023

The manuscript needs revision

thank you 

Reviewer 3 Report

The paper A Threshold Helium Leakage Detection Switch with Ultra Low Power Operation is focusing on Helium gas leakage detection using an electrostatic microelectromechanical system (MEMS) switch. The presented principle uses a dielectric variation causing changes in an electric capacity of a switch. The method is further enhanced by exciting the switch's electrical resonance using proper resistivity and inductivity elements. The authors used the COMSOL Multiphysics finite-element software to model and simulate various scenarios to obtain optimum sensing system.

First, the language level throughout the entire paper is somewhat acceptable, however, I was able to notice various mistakes and errors in the text, which should be corrected. For example: “die-electric” text lines 24, 239, 240, etc., missing Figure number text lines 123, 143, 148, wrong dielectric constant value line 218, etc. I recommend to perform proofreading and corrections in the text.

Next, although section 1. Introduction brings examples of various existing methods for Helium detection, their accuracy and direct comparison with the proposed technique is missing.

In my opinion, authors should elaborate on the temperature dependency of their proposed technique. As the dielectric characteristics and gas parameters are generally temperature dependent, I recommend to include temperature dependency simulations of their presented method.

The language level throughout the entire paper is somewhat acceptable, however, I was able to notice various mistakes and errors in the text, which should be corrected. For example: “die-electric” text lines 24, 239, 240, etc., missing Figure number text lines 123, 143, 148, wrong dielectric constant value line 218, etc. I recommend to perform proofreading and corrections in the text.

Round 2

Reviewer 3 Report

All my questions and comments were sufficiently addressed, I recommend to accept the paper.